# Prior Social Contact and Mental Health Trajectories during COVID-19: Neighborhood *Friendship* Protects Vulnerable Older Adults

**DOI:** 10.3390/ijerph18199999

**Published:** 2021-09-23

**Authors:** Daniel R. Y. Gan, John R. Best

**Affiliations:** 1Gerontology Research Centre, Simon Fraser University, Vancouver, BC V6N 5K3, Canada; john_best@sfu.ca; 2Department of Psychiatry, University of British Columbia, Vancouver, BC V6T 2A1, Canada

**Keywords:** social isolation, emergency preparedness, social support, structural equation modeling, cohesion, neighborhood

## Abstract

Social networking protects mental health during a crisis. Prior contact with social organizations, friends, and non-friend neighbors may be associated with better trajectories of loneliness, depression and subjective memory during COVID-19. Regression analysis was conducted using longitudinal data from a representative sample of *n* = 3105 US adults aged ≥55 in April–October 2020. Latent profile analysis was also conducted. Prior contact with friends (B = −0.075, *p* < 0.001), neighbors (B = −0.048, *p* = 0.007), and social organizations (B = −0.073, *p* < 0.001) predicted for better mental health during COVID-19. Three profiles were identified: Profile 1 had the best outcomes, with prior contact with social organizations (B = −0.052, *p* = 0.044) predicting decreasing loneliness. For Profile 2, prior ‘meeting’ contact with friends predicted decreasing loneliness (B = −0.075, *p* < 0.001) and better subjective memory (B = −0.130, *p* = 0.011). Conversely, prior contact with neighbors (B = −0.165, *p* = 0.010) predicted worsening loneliness among Profile 3. The COVID-19 pandemic has had a differential impact on the mental health trajectories of aging adults with social ties of different strengths. Stronger neighborhood networks are important to mitigate poor mental health outcomes among vulnerable older adults during a crisis. Older adults who are living alone and had relied on non-friend neighbors for social connectedness require additional community supports. Policy interventions are required to mitigate the mental health impact of future pandemics.

## 1. Introduction

The COVID-19 has impacted many aspects of our lives including our social lives [1]. Various countries responded with school closures and/or mobility restrictions, some in a more timely manner than others [2]. In the United States, varying degrees of mobility restrictions were implemented across different states [3], making it an interesting case for the study of COVID-19 effects on older adults’ health and well-being. While some populations re-engaged in social activities “per the old norms” which led to over 10,000 cases a day (e.g., in Florida), other populations “pulled back their opening plans” (e.g., Texas) [1] (p. 3).

For many older adults, social contact has been reduced amid the COVID-19 pandemic to minimize transmission risks [4]. Reduced social contact may have had an impact on specific aspects of mental health [5]. Enhancing mental health through social contact while distancing becomes a priority as some older adults continue to shelter [6]. Yet, the needs may be beyond the current level of resources. Little is known of whom such interventions should target moving forward [7,8]. For instance, some studies suggest that women may be especially vulnerable [7], whereas others highlight the needs of people with cognitive impairment who are living alone [8].

It is also unclear whether—and under what circumstances might—social contact prior to an emergency generates sufficient social capital so as to buffer individuals against its impact [9]. Amid situations of widespread powerlessness, some individuals (“natural neighbors”) are known to step up to support their immediate communities [10] (p. 146). Such acts of neighboring were observed in the UK as many contacted neighbors to check if they needed help with food shopping and chatted with them face-to-face at a safe distance [11]. COVID-19 provided a “pretext” or justified helpful neighborly interactions [12]. These informal assistance networks may be reactivated to alleviate mental distress in the event of future recurrences of COVID-19.

The idea that communities could become responsible enough to take care of their own first caught the attention of policymakers in the 1970s [13,14]. This “third way” of social service delivery through grassroots organizing [15] is enticing because it potentially shifts the “burden” of care away from social policy and saves public monies [16] (p. 2139). The rise of apps for neighborhood social networks like Nextdoor in the UK and USA during COVID-19 [17] pointed to its plausibility but also showed that structures for inclusion need to be carefully orchestrated, especially in the presence of stark societal fault lines [18]. Neighborhoods of lower socio-economic statuses should not be left to suffer the disproportionate impacts of future pandemics [19,20,21].

From the perspective of asset-based community development (ABCD), “service innovation” could aim to serve individuals of greatest need by complementing grassroots organizing [22]. Because loneliness stems from a negative appraisal of one’s social contact [23], it is suggested that “older people *who previously had not reported being socially isolated and lonely* may be disproportionately affected by the requirements of social isolation due to COVID-19, because of the removal of social contacts, which may have occurred during grocery shopping, attending community groups and places of worship and other day-to-day activities” [24] (p. 2044, emphasis added).

In other words, older adults who had been homebound could potentially demonstrate greater resilience in the face of mobility restrictions. The mental health effects of COVID-19 are not straightforward and may vary greatly from individual to individual, just as the salubrious effects of neighborhood social contact vary from context to context [25]. Given the uniqueness of individual responses to local and personal contexts, it may be more helpful to tailor (tele)health interventions to individuals and their contexts based on their mental health outcomes than sociodemographic variables alone.

### Aims and Hypotheses

This paper demonstrates how specific older adults and neighborhoods that could benefit from neighborhood-based telehealth or other mental health interventions may be identified through an application of latent profile modeling with continuous distal outcomes [26]. Depending on the strength of relationships in available social circles and the ease of accessing resources in these relationships, individuals may develop different ways of coping. Some may exhibit poorer mental health than others. Thus, we hypothesize that (1) prior social contact may affect mental health trajectories during COVID-19.

Given that loneliness and depression may affect neural networks [27], some ways of coping may shape personalities over time. Some older adults may experience persistent loneliness whereas others may improve over time. Thus, we hypothesize that (2) there are different profiles of mental health trajectories during COVID-19.

As an indicator of neighborhood social capital, prior contact with neighbors may result in neighborly support that could have improved mental health trajectories during COVID-19. Amid voluntary or imposed mobility restrictions, neighborhood social capital may be crucial. As such, we hypothesize that (3) controlling for prior contact with others, prior contact with neighbors is associated with better mental health trajectories during COVID-19.

## 2. Materials and Methods

Data were collected from online questionnaires completed monthly from April through October 2020 (7 months) of the COVID-19 Coping Study [28]. Participants are a representative sample of community-dwelling adults aged 55 and above from all 50 US states and the District of Columbia. Participants were recruited using internet-based quota sampling [28]. Monthly follow-up questionnaires assessed mental health and well-being using standardized surveys and open-ended questions. Eligible participants for this study are participants with ≥3 follow-ups after baseline.

Social contact prior to COVID-19 was measured at baseline. Participation in social organizations was measured as a single item with responses ranging from 0 (Less than once a month) to 4 (Daily or almost daily). In-person social contact with friends was measured using an item with responses ranging from 0 (Less than once a month) to 3 (Three or more times a week); participants who indicated having no friends in a previous question were recoded as −1. A single-item measure of tele-conversation with friends was included with the same response options. Social contact with neighbors who were not considered friends was measured as a single item with responses ranging from 0 (Less than once a month) to 3 (Three or more times a week).

Self-rated memory was measured using an item with five options ranging from 0 (Poor) to 4 (Excellent). Loneliness was measured using the 3-item UCLA Loneliness scale with three options ranging from 1 (Hardly ever) to 3 (Often); scores were summed [29]. Depressive symptoms were measured using the 8-item CES Depression scale with two options: 0 (No) and 1 (Yes); scores were summed [30].

We controlled for age, sex, education at baseline, and whether participants live alone. In addition, we also controlled for whether participants receive regular in-home assistance or care as an indicator of health status. We did not control for race, as its influence is among the subject of subsequent qualitative investigations, e.g., living arrangements and social networks, for context-sensitive interventions [31].

Individual-level random intercepts and slopes in self-rated memory, loneliness, and depression were estimated using linear mixed models with restricted maximum likelihood in R v4.0.3. This approach does not require the explicit imputation of missing data to produce unbiased estimates of intercepts and slopes under the assumption data are missing at random [32].

Regression analyses were conducted in Stata v15.0 to examine associations between outcome and exposure variables in the full sample and in each profile. An index of intra-individual mean values of outcome variables was created for the full sample, and mental health was examined in relation to each of social contact with organizations, friends and neighbors.

Latent profile analysis (LPA) was conducted on random intercept and slope values to identify profiles of mental health trajectories during COVID-19. LPA solutions with two to four distinct profiles were considered. In addition to traditional measures of model fit including AIC and BIC, interpretability and size of the resulting profiles guided the selection of the best LPA model. Participants were assigned to a specific profile based on their maximum posterior probabilities. Profile characteristics were examined using multinomial logistic regression. Slopes of memory, loneliness, and depression were examined in relation to exposure variables in separate regression models for each profile.

## 3. Results

### 3.1. Sample Characteristics

3105 participants are included in this study. The mean age of the sample is 67.5 years (SD = 7.3). 71.0% of the sample were female. Mean level of education was 3.3 (undergraduate degree; SD = 0.8, Range = 0–4). 26.6% of the sample reported living alone. 5.8% reported receiving regular in-home assistance.

### 3.2. Regression Analysis

As shown in Table 1, prior contact with friends (B = −0.075, *p* < 0.001) and organizations (B = −0.073, *p* < 0.001) were most strongly associated with better mental health during COVID-19 in adjusted models of the full sample. Prior contact with neighbors (B = −0.048, *p* = 0.007) was also associated with better mental health during COVID-19.

### 3.3. Latent Profile Analysis

As shown in Table 2, a three-profile solution provided the best fit to the data according to AIC and BIC values. Visually, plots of outcome means over time (not shown) indicate declining memory and increasing loneliness for Profiles 2 and 3, and gently decreasing depressive symptoms for Profile 3. Variation in the intercept exceeded variation in the slope based on the initial linear mixed models fitted to loneliness, depressive symptoms and subjective memory. Intercept, slope and residual variances are 2.10, 0.02, and 0.82 respectively for loneliness, 3.77, 0.04, and 1.48 respectively for depressive symptoms, and 0.65, 0.005, and 0.19 for subjective memory.

Profile 1 (*n* = 2014) had the best outcomes. Compared to individuals in Profile 2, individuals in Profile 1 were older (C = 0.026, 95% CI = 0.013 to 0.039, *p* < 0.001), more likely to be males (C = −0.386, 95% CI = −0.590 to −0.183, *p* < 0.001), less likely to live alone (C = −0.632, 95% CI = −0.827 to −0.438), and participated in social organizations more frequently (C = 0.102, 95% CI = 0.028 to 0.177, *p* = 0.007) prior to COVID-19.

Among individuals in Profile 1, prior contact with social organizations was associated less depression (B = −0.065, *p* = 0.008) and better self-rated memory (B = 0.065, *p* = 0.008) during COVID-19. At the same time, prior contact with social organizations was associated with decreasing loneliness (B = −0.069, *p* = 0.006) during COVID-19. See Table 3.

Profile 2 (*n* = 774) had average outcomes. Among individuals in Profile 2, prior contact with neighbors was associated with less depressive symptoms (B = −0.102, *p* = 0.007) and better self-rated memory (B = 0.078, *p* = 0.037) during COVID-19. At the same time, prior contact with friends was associated with decreasing loneliness (B = −0.107, *p* = 0.017) and increasing self-rated memory (B = −0.107, *p* = 0.015) during COVID-19. See Table 4.

Profile 3 (*n* = 317) had the worst outcomes. Compared to individuals in Profile 2, individuals in Profile 3 were younger (C = −0.035, 95% CI = −0.056 to −0.014, *p* < 0.001), less educated (C = −0.268, 95% CI = −0.425 to −0.110, *p* = 0.001), more likely to live alone (C = 0.663, 95% CI = 0.379 to 0.946, *p* < 0.001), and had less frequent contact with friends (C = −0.247, 95% CI = −0.411 to −0.083, *p* = 0.003) prior to COVID-19. Among individuals in Profile 3, prior contact with neighbors was associated with increasing loneliness during COVID-19 (B = 0.195, *p* = 0.001). See Table 5.

## 4. Discussion

Consistent with concepts such as social integration and the natural neighborhood network [33,34], the current study observed that stronger social networks were associated with better mental health during COVID-19 [35,36]. Social contact with neighbors had a weaker but nonetheless statistically significant association with mental health during COVID-19 [37].

Latent profile analysis showed that there are three distinct profiles with significant differences in sociodemographic characteristics, mental health trajectories, and responses to social contact. Most older adults (Profile 1) had participated frequently in social organizations. Networks that were developed within social organizations likely eased depression and maintained their subjective memory. This is consistent with findings that older ages were more likely to provide and receive support during COVID-19, which were associated with better mental health [38]. Social organizations may be the primary sources of this mutual support during COVID-19.

Some older adults (Profile 2) participated in social organizations less frequently but successfully relied on neighbors for improved mental health outcomes. These individuals benefited from prior contact with friends, which suggests that some neighbors may have become friends over time. Studying the formation of neighborhood friendships may further elucidate the experiences of these older adults [39]. In this regard, our study showed that contact with friends through phones alone is not associated with better mental health. Friendship requires face-to-face meetings, much in the same way that mutual aid or self-help groups have operated [13] (p. 136). Face-to-face meetings may be crucial to sustaining the mutuality of the friendship.

Of greatest concern is approximately 10% of the older adults (Profile 3) for whom mental health remained poor throughout seven months of COVID-19. Many of these adults live alone and may have relied on neighbors for their social network and mental wellbeing prior to COVID-19. However, their relationship with their neighbors had not developed into friendship [40]. As such, their social contact during COVID-19 would likely be curtailed. It is less likely that casual acquaintances in the neighborhood had checked in on them [11], which may explain why prior contact with neighbors was associated with increasing loneliness for these older adults [24]. This is consistent with findings of poorer mental health among those who increased social contact only during COVID-19 [35].

Our findings support the hypothesis of Brooke and Jackson [24] that some older adults who had relied on casual contact for a sense of social connectedness and mental health are likely most affected by COVID-19. As a form of emergency preparedness, post-pandemic community development could thus focus on providing opportunities to form friendships in the neighborhood [39], especially for older adults who do not participate in social organizations. Neighborhood organizations should be supported to reach out (e.g., via telephone) to older adults who are not active members of the social network in the neighborhood [6,41]. City officials and social policymakers should ensure that neighborhood organizations are sufficiently resourced to carry out these critical outreaches to protect the mental health of vulnerable older adults, especially those in neighborhoods of lower socioeconomic status, and those without internet access from home [22,42].

### Limitations

Despite the longitudinal study design which could support causal theorization for the advancement of neighborhood effects, bi-directional causality remains possible [43]. The variation in changes in our primary measures—memory, loneliness, and depression—was small and much less than the variation in the intercept values. As a result, the latent profiles are primarily distinguished by the variation in the intercept values. Whether the intercept variations reflect stable differences in memory, depression and loneliness that would remain after the pandemic is beyond the scope of this paper. Longer follow-ups are required. Greater representations of males and those with lower education (e.g., via other data collection methods) may be helpful [44,45,46]. Social contact during COVID-19, including neighborhood friendship and support from family, may be included as explanatory variables for path analysis.

Although quantitative data analysis involving latent profile analysis may be useful to identify and characterize the networks of vulnerable older adults for targeted interventions, additional qualitative study is required to provide a richer understanding of older adults networks. Future research may seek to understand plausible mechanisms by examining qualitative data of participants’ lived experiences and comparing across profiles. Community-level structural variables (e.g., neighborhood socioeconomic status) may alter individual experiences of the COVID-19 syndemic pandemic [20,47,48,49]. Qualitative methods are appropriate to “understand social processes or social structures … the settings, groups or individuals … which cannot be pre-selected” otherwise [50].

## 5. Conclusions

This study used longitudinal data to show that prior contact with organizations, friends and neighbors is important for the mental health of older adults during COVID-19. Approximately one in ten older adults had poorer social networks and relied on non-friend neighbors for a general, diffused sense of social connectedness and mental health. These weak relationships are insufficient to overcome mental health challenges in the event of a pandemic. They may be “too little, too late.” More robust community development work is required to ensure that these older adults are included in “natural neighborhood networks” for their well-being [38].

## Figures and Tables

**Table 1 ijerph-18-09999-t001:** Regression on an index of mental health in the full sample (*n* = 3033) ^1^.

	Beta	*p* > |t|
**Control variables**		
Age	−0.089	**0.000**
Sex (Female = 1)	0.061	**0.001**
Education	−0.125	**0.000**
Lives alone	0.202	**0.000**
Home assistance	0.045	**0.010**
**Exposure variables**		
Friends meet	−0.075	**0.000**
Friends phone	−0.010	0.628
Neighbors	−0.048	**0.007**
Organizations	−0.073	**0.000**
Constant		0.000

^1^ Adj. R-squared = 0.099.

**Table 2 ijerph-18-09999-t002:** Fit statistics of various solutions from latent profile analysis (LPA) ^1^.

Solutions	Akaike’s Information Criterion (AIC)	Bayesian Information Criterion (BIC)
2-profile	6451.594	6566.369
3-profile	5623.557	5780.617
4-profile	5633.183	5820.447

^1^ Lower values indicate a better fit.

**Table 3 ijerph-18-09999-t003:** Regression analysis on outcome variables among individuals in Profile 1 (best outcomes). *n* = 1969.

	Loneliness Intercept		Loneliness Slope		Depression Intercept		Depression Slope		Memory Intercept		Memory Slope	
	Beta	*p* > |t|	Beta	*p* > |t|	Beta	*p* > |t|	Beta	*p* > |t|	Beta	*p* > |t|	Beta	*p* > |t|
**Control variables**												
Age	−0.035	0.135	0.073	**0.002**	−0.050	**0.034**	0.053	**0.027**	−0.102	**0.000**	0.022	0.368
Sex (Female = 1)	0.067	**0.004**	0.016	0.511	0.167	**0.000**	−0.009	0.709	0.030	0.195	−0.089	**0.000**
Education	0.014	0.530	0.000	0.997	0.000	0.992	0.020	0.371	0.137	**0.000**	−0.011	0.632
Lives alone	0.180	**0.000**	−0.077	**0.001**	−0.005	0.815	−0.059	**0.011**	0.024	0.300	0.011	0.646
Home assistance	−0.010	0.672	0.044	0.052	0.041	0.070	0.057	**0.012**	−0.013	0.574	−0.003	0.885
**Exposure variables**												
Friends meet	0.004	0.891	−0.026	0.346	0.013	0.634	0.053	0.053	0.029	0.285	−0.021	0.437
Friends phone	−0.012	0.656	0.011	0.674	0.026	0.308	−0.017	0.519	0.051	0.051	0.006	0.806
Neighbors	−0.020	0.394	0.017	0.464	0.001	0.970	0.024	0.297	0.003	0.892	−0.019	0.423
Organizations	0.047	0.057	−0.069	**0.006**	−0.065	**0.008**	−0.014	0.582	0.065	**0.008**	0.013	0.588
Constant		0.001		0.002		0.000		0.005		0.113		0.498
Adj. R-squared	0.038		0.012		0.034		0.009		0.038		0.005	

**Table 4 ijerph-18-09999-t004:** Regression analysis on outcome variables among individuals in Profile 2 (average outcomes). *n* = 757.

	Loneliness Intercept		Loneliness Slope		Depression Intercept		Depression Slope		Memory Intercept		Memory Slope	
	Beta	*p* > |t|	Beta	*p* > |t|	Beta	*p* > |t|	Beta	*p* > |t|	Beta	*p* > |t|	Beta	*p* > |t|
**Control variables**												
Age	0.012	0.741	0.008	0.843	−0.001	0.978	0.058	0.132	−0.055	0.151	0.130	**0.001**
Sex (Female = 1)	0.020	0.584	0.027	0.469	0.042	0.258	0.070	0.061	0.037	0.310	−0.095	**0.010**
Education	−0.029	0.426	0.080	**0.030**	−0.036	0.330	0.025	0.495	0.156	**0.000**	0.005	0.886
Lives alone	0.248	**0.000**	−0.027	0.466	−0.021	0.575	−0.063	0.090	0.012	0.751	0.035	0.343
Home assistance	−0.038	0.285	0.036	0.324	0.009	0.809	−0.002	0.964	0.013	0.729	−0.029	0.419
**Exposure variables**												
Friends meet	−0.039	0.364	−0.107	**0.017**	−0.069	0.123	0.005	0.914	−0.035	0.426	−0.107	**0.015**
Friends phone	0.030	0.463	0.040	0.341	−0.011	0.800	−0.040	0.345	0.062	0.135	0.064	0.119
Neighbors	0.034	0.355	0.028	0.466	−0.102	**0.007**	0.041	0.284	0.078	**0.037**	0.032	0.393
Organizations	0.059	0.142	0.053	0.203	0.014	0.728	−0.031	0.451	0.035	0.390	0.021	0.603
Constant		0.258		0.518		0.000		0.073		0.070		0.001
Adj. R-squared	0.062		0.005		0.011		0.001		0.031		0.029	

**Table 5 ijerph-18-09999-t005:** Regression analysis on outcome variables among individuals in Profile 2 (average outcomes). *n* = 757.

	Loneliness Intercept		Loneliness Slope		Depression Intercept		Depression Slope		Memory Intercept		Memory Slope	
	Beta	*p* > |t|	Beta	*p* > |t|	Beta	*p* > |t|	Beta	*p* > |t|	Beta	*p* > |t|	Beta	*p* > |t|
**Control variables**												
Age	−0.004	0.940	−0.002	0.978	−0.057	0.345	−0.043	0.477	0.080	0.170	0.143	**0.018**
Sex (Female = 1)	−0.104	0.067	−0.112	0.051	−0.011	0.846	−0.109	0.063	0.153	**0.007**	−0.115	**0.049**
Education	0.032	0.577	0.018	0.752	−0.055	0.351	0.068	0.250	0.095	0.097	0.024	0.687
Lives alone	0.270	**0.000**	−0.071	0.216	0.035	0.552	−0.012	0.833	−0.019	0.740	0.024	0.679
Home assistance	−0.036	0.526	0.028	0.627	0.060	0.309	−0.012	0.835	−0.077	0.180	−0.009	0.875
**Exposure variables**												
Friends meet	−0.029	0.697	0.019	0.801	−0.075	0.335	0.042	0.582	0.167	**0.027**	−0.066	0.394
Friends phone	0.046	0.524	−0.019	0.790	0.138	0.063	−0.043	0.565	−0.038	0.597	0.006	0.937
Neighbors	−0.019	0.740	0.195	0.001	−0.080	0.177	0.105	0.076	0.096	0.096	−0.037	0.526
Organizations	0.023	0.715	−0.075	0.240	0.039	0.544	0.006	0.924	−0.072	0.253	−0.021	0.744
Constant		0.003		0.672		0.000		0.555		0.000		0.022
Adj. R-squared	0.055		0.032		0.001		0.003		0.060		0.010	

## Data Availability

The authors thank Lindsay Kobayashi and Jessica Finlay for access to data from the COVID-19 Coping Study, and constructive feedback on earlier versions of this article and its methods. Deidentified, non-geographic data may be securely accessed through reasonable request and collaboration with a proposal form from lkob@umich.edu or jmfinlay@umich.edu to ensure non-overlap.

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
