# Peer review of "Prior Social Contact and Mental Health Trajectories during COVID-19: Neighborhood Friendship Protects Vulnerable Older Adults"

_ijerph, 2021, doi:10.3390/ijerph18199999_

Round 1
Reviewer 1 Report
Dear authors,
I find the article of good significance and addressing an important topic, namely the importance or the role of the neighborhood (and social organizations and friends) in caregiving and support to older adults. It confirms other (both quantitative and qualitative) research that the role of the neighborhood/neighbors is sometimes limited, but for some an important source of social contact.
Please find some comments below, that in my opinion can further enrich the article:
- The second part of the title "neighboring protects vulnerable adults" could be adjusted in order to better stress the need to create stronger neighborhood networks
- In the first paragraph of the 'aims and hypotheses' and the paragraph just before it you mentioned "telehealth", but you do not reflect on this later on. "it may be more helpful tailor telehealth interventions to individuals and their contexts bases on their mental health outcomes than sociodemographic variables alone" -> I agree, but do not think telehealth is an alternative to (local - neighborhood-based) social contacts. So either expand the telehealth topic (why and where can it beneficial or complementing local networks (beyond the pandemic)? or use "(tele)health"
- Data/insight about the role of the family is missing. Wasn't this part of the questionnaire? It would be interesting to see what the role is of prior contact with family and in future research if a loss of other networks was compensated by family members
- I think the article can be further enriched by making reference to qualitative work that focused on the neighborhood and neighborhood networks during COVID-19 (e.g. several articles in the same special issue among others).
All the best!
Author Response
IJERPH Response to Reviewers
Dear Reviewers,
Thank you for your helpful feedback. Please see our response in CAPS below.
R1:
I find the article of good significance and addressing an important topic, namely the importance or the role of the neighborhood (and social organizations and friends) in caregiving and support to older adults. It confirms other (both quantitative and qualitative) research that the role of the neighborhood/neighbors is sometimes limited, but for some an important source of social contact.
Please find some comments below, that in my opinion can further enrich the article:
The second part of the title "neighboring protects vulnerable adults" could be adjusted in order to better stress the need to create stronger neighborhood networks
THANK YOU FOR THE KIND WORDS! AMENDED.
In the first paragraph of the 'aims and hypotheses' and the paragraph just before it you mentioned "telehealth", but you do not reflect on this later on. "it may be more helpful tailor telehealth interventions to individuals and their contexts bases on their mental health outcomes than sociodemographic variables alone" -> I agree, but do not think telehealth is an alternative to (local - neighborhood-based) social contacts. So either expand the telehealth topic (why and where can it beneficial or complementing local networks (beyond the pandemic)? or use "(tele)health"
AMENDED.
Data/insight about the role of the family is missing. Wasn't this part of the questionnaire? It would be interesting to see what the role is of prior contact with family and in future research if a loss of other networks was compensated by family members
THANK YOU FOR POINTING OUT THESE IMPORTANT QUESTIONS. GIVEN OUR PUBLIC HEALTH FOCUS ON THE BENEFITS OF NON-FAMILY NETWORKS WHICH IS ARGUABLY MORE MODIFIABLE, WE OPTED NOT TO STUDY FAMILY NETWORKS IN THIS PAPER. INSTEAD, WE CONTROLLED FOR LIVING ARRANGEMENT.
I think the article can be further enriched by making reference to qualitative work that focused on the neighborhood and neighborhood networks during COVID-19 (e.g. several articles in the same special issue among others).
THANK YOU FOR THE SUGGESTION. WE ADDED TWO REFERENCES.
R2:
This is an interesting paper based upon a good research design relying upon quantitative methods. The authors refer to future qualitative research, which might indeed enhance the interest and impact of this specific study.
As a scholar with expertise in qualitative rather than quantitative methods, I cannot comment on the scientific soundness of the statistical analysis. I would like to comment however on some limits and constraints that are not explicitly discussed by the authors.
1) This study analyses the results of a longitudinal (7 months) research using online questionnaires. The authors however do not discuss how the choice for this particular research method might affect the outcome. Indeed it is fully conceivable that frail elderly people, those with cognitive impairments and many of those with a lower socio-economic status do not participate in such research because of a lack of familiarity with the internet and online applications. Hence there might be a bias in the results from the very beginning, because people who might conceivably be most vulnerable for negative outcomes are also least likely to actively participate by filling out online questionnaires.
WE ACKNOWLEDGE THE LIMITATIONS OF COLLECTING DATA ONLINE. IN LIGHT OF THE POSSIBLY GREATER VULNERABILITY OF THOSE WITHOUT INTERNET ACCESS, WE ARGUE THAT OUR STUDY PROVIDED A CONSERVATIVE ESTIMATE OF THE PROPORTION OF OLDER ADULTS FOR WHOM MENTAL HEALTH REMAINED POOR (PROFILE 3).
2) I am wondering about the absence of references to family contacts. In Europe it is not only friends and neighbors, but certainly also family networks that play a very important role in older adults' lives. It is strange that this is not spelled out as an important factor in the research.
GIVEN OUR PUBLIC HEALTH FOCUS ON THE BENEFITS OF NON-FAMILY NETWORKS WHICH IS ARGUABLY MORE MODIFIABLE, WE OPTED NOT TO STUDY FAMILY NETWORKS IN THIS PAPER. INSTEAD, WE CONTROLLED FOR LIVING ARRANGEMENT.
3) It is a bit odd that neighborhood characteristics are not seen as important factors in a study that aims to analyze how neighborly contacts play a role in mental health trajectories during COVID-19. Neither the socio-economic status nor the density or morphology (urban, suburban, rural, apartment buildings, town houses, ...) are discussed. I assume that the authors do not have access to such data, and hence cannot take them into account in their analysis. They should however mention this, because it is very well possible that also neighborhood characteristics might be co-responsible for the effects that become visible in this study.
THAT IS RIGHT. WE INCLUDED THIS AS A LIMITATION.
I find the paper valuable and would suggest that the authors add a paragraph on the limits and constraints of the current study, explaining how they might be addressed in future research.
AMENDED.
R3:
I believe that the introduction needs to be more structured. It is necessary to show the results of the various studies that show the impact on mental health of all the restrictions resulting from the pandemic (confinement, break with routine, loss of employment, economic problems, etc.). All these aspects can be related to the mental health of older adults.
It is also important to refer to studies that have found how loneliness during confinement has affected mental health in older people.
GIVEN THE MANY PAPERS THAT ARE BEING PUBLISHED ON THE MENTAL HEALTH IMPACT OF VARIOUS CHANGES DURING COVID-19 (AND THE WORD LIMIT), WE OPTED TO PROVIDE A BRIEF REFERENCE TO THESE CHANGES [1], AND TO FOCUS SOLELY ON THE MENTAL HEALTH IMPACT OF SOCIAL CONTACT WITH ORGANIZATIONS, FRIENDS, AND NEIGHBORS, WHICH MAY BE MORE EASILY MODIFIABLE THROUGH POLICIES AND ARE GENERALLY RELEVANT TO OLDER ADULT POPULATIONS.
WE BRIEFLY DISCUSSED LONELINESS IN THE INTRODUCTION (5TH PARAGRAPH), AND ADDED A REFERENCE ON LONELINESS IN THE DISCUSSION.
I think it is important to differentiate between a 55 year old adult and a 75 year old adult. Personal and social characteristics can be very different at 55 and 75.
WE AGREE WITH THE SIGNIFICANCE OF AGE 70, WHICH WE HAVE FOUND IN A SEPARATE STUDY TO BE THE AGE AROUND WHICH DEPRESSIVE SYMPTOMS INCREASE IN NORTH AMERICA (BEST, GAN, ET AL., 2021; JOURNAL OF AFFECTIVE DISORDERS). FOR THE PURPOSES OF IDENTIFYING LATENT PROFILES BASED ON MENTAL HEALTH VARIABLES, WE OPTED TO CONTROL FOR AGE AS AN ORDINAL VARIABLE INSTEAD OF STRATIFIYING BY DICHOTOMIZED AGE GROUPS (I.E., YOUNG-OLD VS. OLD-OLD).
It is not clear how memory was measured. Mental health is a broad concept that does not only refer to depression and loneliness. I think it would have been better to include the measurement of other variables such as anxiety, coping with stress, etc.
I think it is important to include reliability data on the measurement scales used in this study.
PER PARA 3 OF THE METHODS SECTION, WE USED A SINGLE-ITEM MEASURE OF SUBJECTIVE MEMORY. WE PROVIDED REFERENCES TO THESE COMMONLY-USED MEASURES WHERE AVAILABLE.
GIVEN OUR INTEREST IN CHANGES TO PERSONALITY DURING COVID-19 (HYPOTHESIS 2), WE OPTED TO STUDY MORE STABLE “TRAITS” SUCH AS DEPRESSION AND LONELINESS INSTEAD OF “STATES.” WE NOTE THAT THESE BROAD CATEGORIES MAY NOT ALWAYS BE HELPFUL.
The characteristics of the sample are usually presented in the "method" section, not in the results.
WE NOTE POSSIBLE DIFFERENCES IN DISCIPLINARY(?) NORMS. WE OPTED TO INCLUDE SAMPLE CHARACTERISTICS IN THE RESULTS SECTION BASED ON GENERAL PRACTICES IN THE PAPERS CITED.
It would be interesting to better specify descriptive data. For example, "the average educational level of the sample was 3.3", specify what this means.
THANK YOU FOR POINTING THIS OUT. WE AMENDED ACCORDINGLY.
It could be pointed out if and how gender has been taken into account. 71% were women.
WE ACKNOWLEDGED THIS DATA LIMITATION IN SECTION 3.1.
I consider it necessary to include a specific section on limitations, indicating aspects that have not been taken into account in this study.
AMENDED.
Reviewer 2 Report
This is an interesting paper based upon a good research design relying upon quantitative methods. The authors refer to future qualitative research, which might indeed enhance the interest and impact of this specific study.
As a scholar with expertise in qualitative rather than quantitative methods, I cannot comment on the scientific soundness of the statistical analysis. I would like to comment however on some limits and constraints that are not explicitly discussed by the authors.
1) This study analyses the results of a longitudinal (7 months) research using online questionnaires. The authors however do not discuss how the choice for this particular research method might affect the outcome. Indeed it is fully conceivable that frail elderly people, those with cognitive impairments and many of those with a lower socio-economic status do not participate in such research because of a lack of familiarity with the internet and online applications. Hence there might be a bias in the results from the very beginning, because people who might conceivably be most vulnerable for negative outcomes are also least likely to actively participate by filling out online questionnaires.
2) I am wondering about the absence of references to family contacts. In Europe it is not only friends and neighbors, but certainly also family networks that play a very important role in older adults' lives. It is strange that this is not spelled out as an important factor in the research.
3) It is a bit odd that neighborhood characteristics are not seen as important factors in a study that aims to analyze how neighborly contacts play a role in mental health trajectories during COVID-19. Neither the socio-economic status nor the density or morphology (urban, suburban, rural, apartment buildings, town houses, ...) are discussed. I assume that the authors do not have access to such data, and hence cannot take them into account in their analysis. They should however mention this, because it is very well possible that also neighborhood characteristics might be co-responsible for the effects that become visible in this study.
I find the paper valuable and would suggest that the authors add a paragraph on the limits and constraints of the current study, explaining how they might be addressed in future research.
Author Response

(The authors gave the same response as above.)

Reviewer 3 Report
I believe that the introduction needs to be more structured. It is necessary to show the results of the various studies that show the impact on mental health of all the restrictions resulting from the pandemic (confinement, break with routine, loss of employment, economic problems, etc.).
All these aspects can be related to the mental health of older adults.
It is also important to refer to studies that have found how loneliness during confinement has affected mental health in older people.
I think it is important to differentiate between a 55 year old adult and a 75 year old adult. Personal and social characteristics can be very different at 55 and 75.
It is not clear how memory was measured. Mental health is a broad concept that does not only refer to depression and loneliness. I think it would have been better to include the measurement of other variables such as anxiety, coping with stress, etc.
I think it is important to include reliability data on the measurement scales used in this study.
The characteristics of the sample are usually presented in the "method" section, not in the results. It would be interesting to better specify descriptive data. For example, "the average educational level of the sample was 3.3", specify what this means.
It could be pointed out if and how gender has been taken into account. 71% were women.
I consider it necessary to include a specific section on limitations, indicating aspects that have not been taken into account in this study.
Author Response

(The authors gave the same response as above.)

Round 2
Reviewer 3 Report
I still think that there are aspects that can be improved. However, the authors' response seems adequate to me.